# Molecular Profiling of Lymphatic Endothelial Cell Activation In Vitro

**DOI:** 10.3390/ijms242316587

**Published:** 2023-11-22

**Authors:** Marta Turati, Gianluca Mattei, Alessia Boaretto, Alberto Magi, Maura Calvani, Roberto Ronca

**Affiliations:** 1Department of Molecular and Translational Medicine, University of Brescia, 25123 Brescia, Italy; m.turati004@unibs.it; 2Department of Information Engineering, University of Florence, 50139 Florence, Italy; gianluca.mattei@unifi.it (G.M.); alberto.magi@unifi.it (A.M.); 3Department of Pediatric Hematology-Oncology, A. Meyer Children’s Hospital, Scientific Institute for Research, Hospitalisation and Health Care (IRCCS), 50139 Florence, Italy; alessiaboaretto96@gmail.com (A.B.); maura.calvani@meyer.it (M.C.)

**Keywords:** lymphangiogenesis, lymphatic endothelial cells, lymphatic activation, RNA sequencing, molecular profile

## Abstract

The lymphatic vascular system plays a key role in cancer progression. Indeed, the activation of lymphatic endothelial cells (LECs) through the lymphangiogenic process allows for the formation of new lymphatic vessels (LVs) that represent the major route for the dissemination of solid tumors. This process is governed by a plethora of cancer-derived and microevironmental mediators that strictly activate and control specific molecular pathways in LECs. In this work we used an in vitro model of LEC activation to trigger lymphangiogenesis using a mix of recombinant pro-lymphangiogenic factors (VFS) and a co-culture system with human melanoma cells. Both systems efficiently activated LECs, and under these experimental conditions, RNA sequencing was exploited to unveil the transcriptional profile of activated LECs. Our data demonstrate that both recombinant and tumor cell-mediated activation trigger significant molecular pathways associated with endothelial activation, morphogenesis, and cytokine-mediated signaling. In addition, this system provides information on new genes to be further investigated in the lymphangiogenesis process and open the possibility for further exploitation in other tumor contexts where lymphatic dissemination plays a relevant role.

## 1. Introduction

The lymphatic vascular system acquires critical roles in cancer progression, representing the major route for the dissemination of many solid tumors, particularly those of epithelial origin. During tumor progression, cancer cells first spread through the lymphatic vessels (LVs) to the tumor-draining lymph node (dLN), possibly causing the formation of distant organ metastasis, which represents the leading cause of mortality in cancer patients [1].

Initially, tumor lymphogenous dissemination was considered a passive process. However, in recent years, several studies have demonstrated that LVs undergo active modifications in response to specific tumor-derived stimuli, thus facilitating cancer metastasis [2,3]. It has been observed that lymphangiogenic growth factors (e.g., VEGF-C, VEGF-D, PDGF-BB, FGF2, S1P) released by both cancer cells and tumor microenvironment mediate the formation of novel LVs and the dilatation of vessel lumen in different primary tumors [2,4]. This results in an increased surface area of potential interaction between cancer cells and lymphatics, and in an increased flow rate, thus facilitating tumor dissemination [2,5]. Both lymphangiogenesis and lymphatic enlargement are also observed in the dLNs even before the arrival of metastatic cells. The generation of a lymphovascular niche in the dLN, following the expansion of the lymphatic network, promotes cancer dissemination to distant LNs and organs [4,6].

In addition to these remodeling processes, tumor-dependent lymphatic activation may promote molecular interactions between tumor and lymphatic endothelial cells (LECs), thus fostering malignant cell entry into LVs [7]. For example, tumor-secreted VEGF-C and lipoxygenase have been reported to promote the disruption of the lymphatic endothelial barrier, leading to increased permeability and consequent trans-endothelial migration of cancer cells [8,9]. Moreover, lymphatic invasion may also be promoted by the constitutive or induced lymphatic production of soluble factors, mediating tumor cell chemotaxis through the peritumoral LVs [10,11]. Tumor-derived lymphangiogenic factors transported through LVs to the dLN can also affect peripheral LVs, creating a suppressing tumor-immune microenvironment in the dLN [12]. All these observations indicate an active role of both cancer cells and tumor microenvironment in promoting the activation of the lymphatic network to foster cancer progression and metastasis, through the activation of a specific transcriptional profile of lymphatic endothelial cells (LECs) and LVs.

In malignant melanomas, LVs represent the major route for metastatic dissemination, which turns melanoma into a life-threatening tumor, responsible for 90% of skin cancer-related deaths each year [13,14]. Accordingly, a positive correlation has been observed in melanoma between the expression of lymphangiogenic factors at the primary site, LV density, and lymphatic metastasis [15,16].

Despite the widely recognized pivotal role of the lymphatic system in melanoma progression, little is known about the molecular changes that occur in cancer-associated LVs, resulting in the acquisition of tumor-promoting features. In this work, we exploited RNA sequencing to compare the transcriptional profile of primary human lymphatic endothelial cells (HDLECs) treated with a prototypic mix (VFS) of lymphangiogenic stimuli (i.e., VEGF-A, FGF2, and S1P) [17,18], or co-cultured with melanoma cells. This analysis highlights several pathways, mainly involved in vessel development, immune response, and cytokine signaling, that are commonly regulated in both tumor- and VFS-activated HDLECs, thus evidencing general common molecular features in these in vitro modes of lymphatic activation.

## 2. Results

### 2.1. In Vitro Activation of HDLECs and Preparation of RNAseq Samples

Lymphangiogenic activation is a key step in the formation of new LVs under pro-lymphangiogenic stimuli. In vitro, a potent activator of LECs is represented by the pro-lymphangiogenic mixture VEGF-A (40 ng/mL), FGF-2 (40 ng/mL), and S1P (2 µM), also named VFS [17,18]. Here, we verified that treatment of HDLECs with VFS promotes cell activities involved in lymphatic activation, thus allowing us to mimic/recapitulate this process in vitro. As shown in Figure 1A, 24 h of treatment with VFS caused a significant increase in HDLEC proliferation compared with the control. Also, in a 3-dimensional endothelial cell sprouting assay, HDLECs showed significant activation in terms of sprouting capacity when embedded as spheroids in a fibrin gel and treated with VFS (Figure 1B). Also, in order to evaluate the activation potential of melanoma cells on lymphatic cells, HDLECs were co-cultured with A375 human melanoma cells using a transwell apparatus, and the impact of A375 stimulation on HDLECs was assessed, quantifying cell proliferation. As shown in Figure 1C, HDLEC proliferation was significantly increased in the presence of melanoma cells compared to control.

These propaedeutic assays confirmed the capacity of VFS to trigger LECs’ activation in vitro and prompted us to evaluate the transcriptional pathways activated in these cells under VFS-activated conditions. Moreover, given the fact that lymphangiogenesis occurs in cancer in response to pro-lymphangiogenic stimuli produced by tumor cells in order to activate vicinal LECs in a pro-tumor way, we wanted to verify if VFS activation mimics or has points of contact with tumor-associated lymphangiogenesis. For this purpose, HDLECs were treated with VFS mixture or co-cultured with human A375 melanoma cells for 24 h; then, mRNA was purified from HDLECs, and the gene expression profile was analyzed by RNA sequencing (Figure 1D).

### 2.2. Transcriptional Comparison of HDLEC Activation by VFS and Human Melanoma Cells

As shown in the Volcano plot (Figure 2A), a consistent number of genes were significantly modulated under these culture conditions. The analysis in the Venn diagrams (Figure 2B) revealed that 679 statistically significant differentially expressed genes (DEG, adjusted *p*-value ≤ 0.05) were upregulated in HDLECs when stimulated with VFS, while HDLEC co-culture with melanoma cells resulted in the upregulation of 799 genes. Notably, as shown in the Venn diagram, 245 DEG were commonly upregulated in the two culture conditions. Similarly, 442 DEG were significantly downregulated in HDLECs when stimulated with FVS, 500 when HDLECs were co-cultured with melanoma cells, and among these, 175 genes were commonly down-modulated.

As reported more in detail in Appendix A, among the commonly most downregulated genes are the tumor necrosis factor ligand superfamily members *TNFSF18* and *TNFSF15*. TNFSF18 is a cytokine found to be expressed in endothelial cells and is important for the interaction between T lymphocytes and endothelial cells. *TNFSF15*, also called vascular endothelial growth inhibitor (VEGI), is an important anti-angiogenic protein. Also, levels of *CX3CL1*, a chemokine endowed with protective and plasticity properties in synaptic scaling, and of LTB (lymphotoxin beta), an inducer of the inflammatory response system involved in the normal development of lymphoid tissue, are significantly down-modulated. Other highly downregulated genes include GJA4 (encoding for a gap junction protein), the intercellular adhesion molecule 1 *ICAM-1* gene, and the *SERPINE1* gene, (also called PAI-1), a serine protease inhibitor representing the main inhibitor of tissue-type plasminogen activator (tPA) and urokinase (uPA).

The most commonly upregulated genes are reported in Appendix A, and among them, we can find the *GJA3* gene, encoding the Gap junction alpha-3 protein, the G-protein coupled receptor 3 *GPR3* gene, and the Kinesin-like protein KIF11 (encoded by *KIF11* gene), a molecular motor protein essential in mitosis. Moreover, an upregulated expression of *SOCS3*, a gene commonly induced by various cytokines, and playing a relevant role in maintaining endothelial homeostasis and promoting survival, as well as the *DEPP1* gene, which plays a relevant role as an autophagy regulator. Finally, a general upregulation of histone-related genes was found as a common feature in HDLEC stimulation, indicating strong activity in DNA rearrangement and dynamics.

Further, the over-representation analysis (ORA) of the modulated genes allowed for the identification of biological pathways that were significantly and commonly altered in HDLECs under these experimental conditions. As reported in Figure 2C, these genes belong to biological processes that can be attributed to cell activation during lymphangiogenesis or lymphatic activation. Among these, tissue development and morphogenesis genes are significantly activated, as well as cell differentiation and proliferation, tube morphogenesis, and vascular development genes. These are key processes for the activation of LECs and their determination/differentiation to form new lymphatic vessels. Of relevance, wound healing-associated molecular pathways are among the most activated ones, and this is strictly relatable with the need for cells involved in tissue repair to proliferate, migrate, and occupy a wounded area/tissue. Notably, the most altered pathways emerging from the ORA include genes involved in the response to cytokines, and this is extremely reasonable and in line with the key role exerted by cytokines during all the steps of lymphatic activation and determination [2].

Since the in vitro stimulation was made to unveil the level of molecular similarities between HDLEC activation by recombinant proteins (VFS) and melanoma cells, a direct comparison was carried out, evidencing the concordance of genes interrogated in specific molecular pathways. Results of the molecular pathways that were significantly modulated, and the single genes contributing to these pathways are reported in Figure 3, Figure 4 and Figure 5.

As shown in Figure 3, key processes correlated with the activation of the lymphatic network are commonly regulated in the two different stimulatory contexts in vitro. For instance, genes related to vasculature development, hematopoietic or lymphoid organ development, blood vessel development, and circulatory system development share a significant number of genes with concordant modulation.

Figure 4 reports concordant genes involved in wound healing, tube development, tissue morphogenesis, and the regulation of cell differentiation. In these biological processes, cells are forced to activate, migrate, and differentiate, as it happens during LV formation and lymphangiogenesis. Finally, in Figure 5, the concordant and discordant genes related to the response to cytokine, cytokine-mediated signaling, and the regulation of the immune system are reported. Given the central role of the cytokine network in activating LECs during lymphangiogenesis, these genes testify to the relevance and translational validity of this in vitro model.

## 3. Discussion

The activation of LECs in LVs represents a key step in the formation of the tumor-draining lymphatic network. In this context, various cancer and microenvironmental entities play active roles in producing pro-lymphangiogenic mediators. Besides the remodeling processes, tumor-dependent lymphatic activation may promote molecular interactions between tumor and lymphatic endothelial cells, thus fostering malignant cell entry into LVs [7].

All these molecular processes are difficult to monitor in vivo, while in vitro models of lymphangiogenesis have been described and are used to investigate mechanistic activation of LECs and explore potential therapeutic targets. In this study, we attempted to add new value and information to these in vitro models, showing that under the molecular and transcriptional profile, many features and genes can be commonly regulated when we exploit recombinant proteins (VFS) or melanoma cell-derived stimuli as activators of lymphangiogenesis. This is a relevant aspect since, in many cases, recombinant protein mixtures attempt to simulate the complex secretome of tumor cells that pour a plethora of growth factors, including pro-angiogenic and pro-lymphangiogenic effectors, into the tumor microenvironment [2,19,20,21].

Here, we reported that many molecular pathways related to endothelial cell remodeling, activation, and morphogenesis are activated concordantly under VFS and cancer cell-mediated stimulation. In fact, these clusters of genes implicated in macro-processes drive vasculature development, hematopoietic or lymphoid organ development, blood vessels development, wound healing, tube development, tissue morphogenesis, and cell differentiation. Thus, these complex features are nicely recapitulated in vitro not only phenotypically but also under the transcription profile.

Of interest, we found a general upregulation of histone-related genes as a common feature in HDLEC stimulation, suggesting a strong activity in DNA rearrangement and dynamics. Given that the regulation of gene expression through histone modifications in LECs is essential for maintaining proper lymphatic function, including lymphangiogenesis, immune response coordination, and tissue fluid homeostasis, a deeper understanding of the intricate interplay between histone rearrangement and LEC function would be of great interest to unravel the molecular mechanisms that underlie various physiological and pathological processes, including inflammation and cancer metastasis.

Finally, it is worth noting that in vitro activation significantly modulates gene sets involved in the response to cytokines, cytokine-mediated signaling, and regulation of the immune system. These are extremely relevant features in the complex signaling network governing LV formation, immune cell recruitment, and cancer cell motility/plasticity. Just to cite a few examples, it has been reported that CXCL12, highly expressed in tumor-associated LVs but not in normal ones, promotes the dissemination of CXCR4-expressing cancer cells toward distant organs that highly express CXCL12 [15]. Accordingly, tumor-derived VEGF-C has been observed to induce CXCL21 secretion by LVs, thus mediating CCR7-dependent lymphatic invasion of cancer cells [22]. The upregulated lymphatic expression of CXCL21 has also been reported to promote the onset of an immune-tolerant tumor microenvironment, characterized by immunosuppressive cytokines and regulatory cells, thus generating a permissive milieu for cancer progression and metastasis [15]. Also, tumor-derived lymphangiogenic factors transported through the LVs to the dLN can also affect peripheral LVs. Accordingly, it has been reported that VEGF-C-activated lymphatic endothelial cells in the dLN suppress antitumor immunity by scavenging and cross-presenting tumor antigens, leading to T cell dysfunctional activation and deletion in dLNs [12]. In line with the above-mentioned observations, higher expression of lymphangiogenic growth factors, such as VEGF-C or VEGF-D, in the primary tumor correlates with increased metastasis and poor prognosis in several cancer types [5].

Altogether, this “transcriptional picture” of the in vitro lymphangiogenic activation of LECs gives more solidity to this model, unveiling that many key features of this process are recapitulated, thus opening the possibility to further exploit this co-culture condition with other tumor models. Moreover, new genes emerging from this evaluation will be worth deeper investigation in the context of LEC and LV biology both in vitro and in vivo in physiological and tumor-associated contexts.

## 4. Materials and Methods

### 4.1. Cell Lines and Reagents

Human dermal lymphatic endothelial cells (HDLECs, VWR International, Milano, Italy) were maintained in complete EGM-2 medium and treated in EBM medium 2.5% FBS. Human melanoma A375 cells (ATCC, Manassas, VA, USA) were cultured in DMEM supplemented with 10% FBS. DMEM culture medium and trypsin were purchased from GIBCO Life Technologies (Grand Island, NY, USA). EGM-2 and EBM media were from Lonza (Walkersville, MD, USA). Fetal bovine serum (FBS) was provided by Euroclone (Milan, Italy). Recombinant human VEGF (VEGF-A165 isoform) was kindly provided by K. Ballmer-Hofer (PSI, Villigen, Switzerland). Recombinant human FGF2 was obtained from Tecnogen (Caserta, Italy). S1P was purchased from Avanti Polar Lipids Inc. (Alabaster, AL, USA).

### 4.2. Cell Proliferation Assay

HDELC were seeded in culture medium and then treated in EBM medium 2.5% FBS with FGF2 (40 ng/mL), VEGF-A (40 ng/mL), and S1P (2 μM) (FVS). For co-culture experiment, A375 melanoma cells were added to the transwell permeable supports and maintained in culture for the different time points. Then, cells were detached through trypsinization and counted using the MACSQuant Analyzer (Miltenyi Biotec) after 24 and 48 h of incubation.

### 4.3. Spheroid Sprouting Assay

HDLEC spheroids were generated as follows: 600 HDLECs were resuspended in 200 µL of a medium composed by 20% methylcellulose/80% EGM-2 and seeded in 96-well plates with a round bottom. After 18 h, properly formed spheroids were identified and collected. Then, 40 aggregates/experimental points were embedded in a 3-dimensional fibrin gel and treated with the pro-lymphangiogenic cocktail VFS. After 24 h of treatment, sprouts were counted and photographed.

### 4.4. In Vitro HDLEC Stimulation and RNA Extraction

HDLEC were seeded in culture medium and then treated in EBM medium 2.5% FBS with the pro-lymphangiogenic cocktail VFS or co-cultured with melanoma cells (A375) seeded on transwell permeable supports. After 24 h of treatment total RNA was extracted using ReliaPrepTM RNA Cell Miniprep System (Promega).

### 4.5. RNA Sequencing

RNA quantification was carried out using both NanoDrop 2000/2000c spectrophotometers (ThermoFisher) and Qubit RNA High Sensitivity (HS) (ThermoFisher). RNA integrity was assessed using the Agilent RNA 6000 Pico Kit on an Agilent Bioanalyzer, and all samples reported a RNA integrity number (RIN) higher than 9.00. Library preparation was performed using the TruSeq Stranded Total RNA Library Prep Gold Kit (Illumina) according to the manufacturer’s instructions, with an initial input of 300 ng of total RNA. The library quality was assessed using an Agilent DNA 7500 kit on an Agilent Bioanalyzer, followed by library quantification using 1X dsDNA High Sensitivity (HS) (ThermoFisher). Libraries were sequenced on a NextSeq 550 Sequencing System (Illumina) using NextSeq 500/550 v2.5 (150 cycles) High Output Reagent Kits (Illumina). Runs were performed with a 2 × 76 bp configuration. Data from the RNA sequencing have been deposited in the GEO database with the following identification: GSE247178.

### 4.6. In Silico Analysis

Salmon (v. 1.5.1) was used for sequencing and quantification of library transcripts associated with genes utilizing the R package org.Hs.eg.db (v. 3.12.0). The downstream analyses were conducted within the R statistical environment, making use of its relevant packages. Differential expression analysis was executed using DESeq2 (v. 1.30.1), to discern DEG. The comparative investigation was undertaken to evaluate the congruence between activated endothelial cells and co-culture endothelial cells in terms of their pathways. To this end, the ClusterProfiler package (v. 3.18.1) was used to perform ORA, thereby enabling the identification of perturbed pathways in the context of the two cell types compared to the healthy reference. The ORA methodology was applied setting specific thresholds for gene selection, including the adjusted *p*-value (adj. ≤ 0.05) and the absolute log2 fold change (|log2FC| ≥ 1).

### 4.7. Statistical Analyses

Statistical analyses were performed exploiting the statistical package Prism 8 (GraphPad Software Version 8.4.3). Statistical significance was evaluated using an unpaired two-samples *t*-test. For more than two groups of samples, data were analyzed with a 1-way analysis of variance and corrected by the Bonferroni multiple comparison test. ** + *p* < 0.001; # *p* < 0.0001.

## 5. Conclusions

In conclusion, our data demonstrate that in vitro activation of lymphangiogenesis represents a relevant tool to test the effects of both recombinant and tumor cell-derived mediators on LECs. Notably, this protocol can be easily extended to other tumor types where lymphatic activation and dissemination play a relevant role. Indeed, these handy experimental settings can provide useful information on already known or not-yet-characterized pathways and genes, to be further validated in more complex in vitro and in vivo systems.

## Figures and Tables

**Figure 1 ijms-24-16587-f001:**
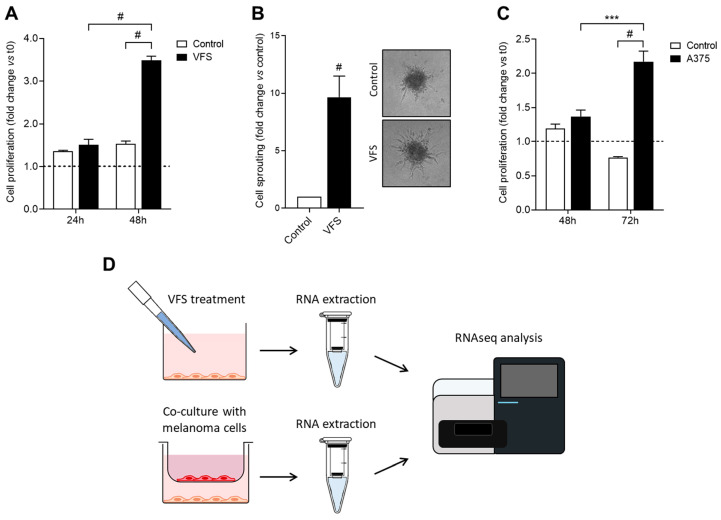
Activating effect of VFS treatment (**A**) on HDLEC proliferation (**B**) and sprout formation. (**C**) Activating effect on HDLECs co-cultured with A375 human melanoma cells. Data are expressed as mean ± SEM (*** *p* < 0.001; # *p* < 0.0001). (**D**) Workflow of the gene expression profile analysis: HDLECs were treated with the pro-lymphangiogenic cocktail VFS or co-cultured with human melanoma A375 cells. After 24 h, total RNA was isolated, and RNA sequencing analysis was performed.

**Figure 2 ijms-24-16587-f002:**
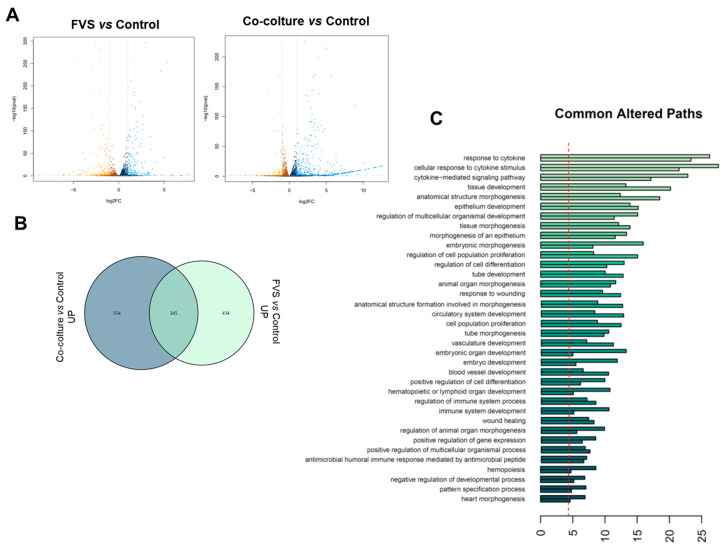
RNAseq was performed on HDLEC stimulated with VFS or co-cultured with melanoma cells, and modulated genes are reported in the Volcano plot, (**A**) commonly altered genes are represented in Venn diagrams, (**B**) and altered pathways identified in the over-representation analysis (ORA) are reported in (**C**).

**Figure 3 ijms-24-16587-f003:**
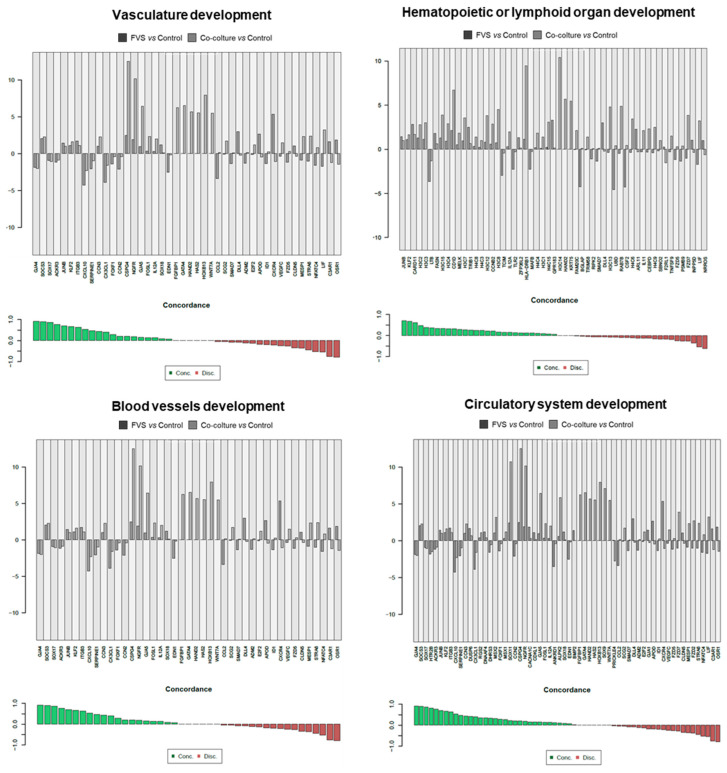
Detail of genes concordantly or discordantly contributing to the molecular pathways that were significantly modulated in the two experimental conditions (VFS or co-culture stimulation of HDLECs).

**Figure 4 ijms-24-16587-f004:**
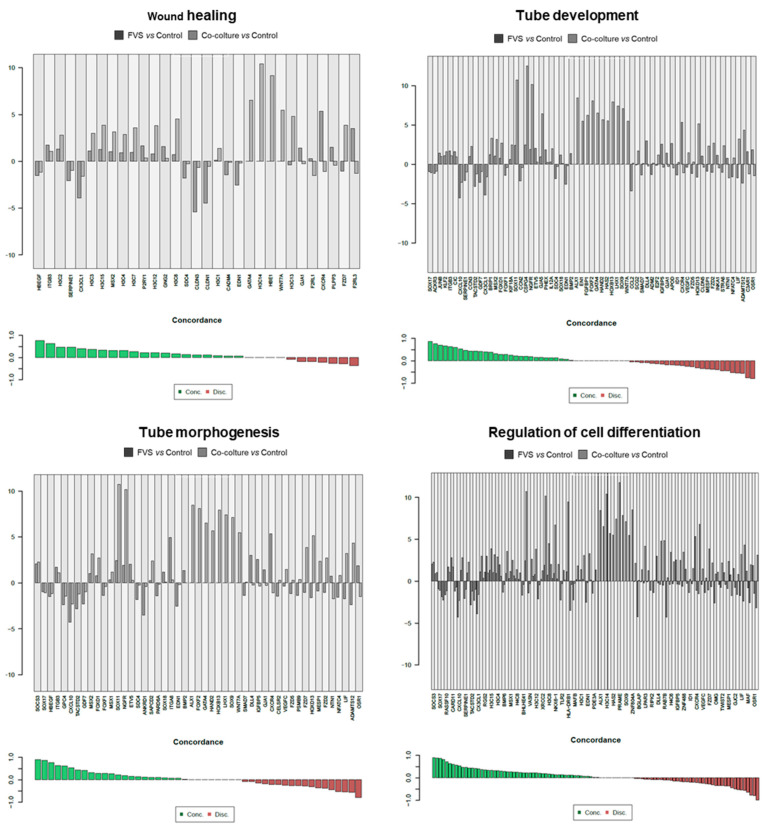
Detail of genes concordantly or discordantly contributing to the molecular pathways that were significantly modulated in the two experimental conditions (VFS or co-culture stimulation of HDLECs).

**Figure 5 ijms-24-16587-f005:**
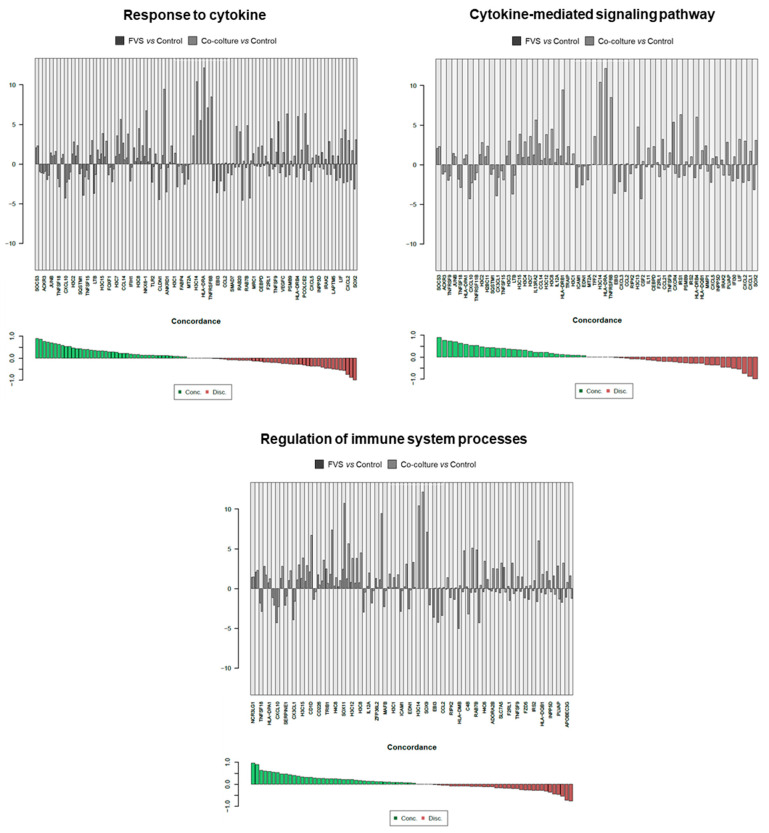
Detail of genes concordantly or discordantly contributing to the molecular pathways that were significantly modulated in the two experimental conditions (VFS or co-culture stimulation of HDLECs).

## Data Availability

Data are contained within the article and Appendix A.

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
