# Peer review of "Molecular Profiling of Lymphatic Endothelial Cell Activation In Vitro"

_ijms, 2023, doi:10.3390/ijms242316587_

Round 1

Reviewer 1 Report

Comments and Suggestions for Authors

The reviewer thinks the aim of study is attractive, methods is appropriate and adequate, results and discussion are reasonable. The reviewer could find almost no concerns in the study. The reviewer recommends to being published after minor revisions.

Figure 1C RNA extraction
The figure made an impression to the reviewer at first sight that the two samples were mixed in RNA extraction. Some modification might be better.

Figure 2

The reviewer thinks “Ctrl” needs its formal term in the figure legends.

Figure 3-5

Name of genes are too small for us readers to read because of low resolution.

Author Response

The reviewer thinks the aim of study is attractive, methods is appropriate and adequate, results and discussion are reasonable. The reviewer could find almost no concerns in the study. The reviewer recommends to being published after minor revisions.

Figure 1C. RNA extraction. The figure made an impression to the reviewer at first sight that the two samples were mixed in RNA extraction. Some modification might be better.

We thank the Reviewer for this suggestion and we have now modified the Figure 1D accordingly.

Figure 2. The reviewer thinks “Ctrl” needs its formal term in the figure legends.

This has been amended in the revised version

Figure 3-5. Name of genes are too small for us readers to read because of low resolution.

This has been amended in the revised version. The resolution has been implemented so that readers can now zoom in.

Reviewer 2 Report

Comments and Suggestions for Authors

Authors analyzed the molecular profiling of lymphatic endothelial cells activation in vitro. They activated the lymphatic endothelial cells via VFS. Then, they performed a co-culture of those cells with human melanoma cells and carried out the RNA sequencing. The design and analyze could be improved.

Lymphatic endothelial cells (LECs) were treated with EBM, activated by VFS, or co-cultured with human melanoma cells. The influence of melanoma cells to LECs must be assessed.

The identified DEGs should be validated by further experiments.

Comments on the Quality of English Language

N/A.

Author Response

Authors analyzed the molecular profiling of lymphatic endothelial cells activation in vitro. They activated the lymphatic endothelial cells via VFS. Then, they performed a co-culture of those cells with human melanoma cells and carried out the RNA sequencing. The design and analyze could be improved.

Lymphatic endothelial cells (LECs) were treated with EBM, activated by VFS, or co-cultured with human melanoma cells. The influence of melanoma cells to LECs must be assessed.

The identified DEGs should be validated by further experiments.

We thank the Reviewer for these suggestions. The aim of our study was to give a preliminary idea that the molecular signature modifications triggered in LECs by canonical recombinant mixtures could be recapitulated by co-culture conditions with tumor cells and vice versa.

We are aware of the fact that the system can be further implemented and validated, and this is part of more complex studies that are actually ongoing to extend these observations.

Due to the scarcity of material available at the moment and the extremely wide and general expression panel presented we are unable to perform single validations, but these have been reported in other studies with single VFS stimulation. However, following the suggestions we have added an in vitro cell proliferation study performed with HDLECs co-cultured with human melanoma cells in transwell apparatus.

We hope that this can further support the validity of our study.

Round 2

Reviewer 2 Report

Comments and Suggestions for Authors

Authors had addressed my concerns.

Author Response

Thank you for your positive comment.